# Rethinking Graph Transformers with Spectral Attention

**Devin Kreuzer** *
McGill University, Mila
Montreal, Canada
devin.kreuzer@mail.mcgill.ca

**Dominique Beaini** *
Valence Discovery
Montreal, Canada
dominique@valencediscovery.com

**William L. Hamilton**
McGill University, Mila
Montreal, Canada
wlh@cs.mcgill.ca

**Vincent Létourneau**
University of Ottawa
Ottawa, Canada
vletour2@uottawa.ca

**Prudencio Tossou**
Valence Discovery
Montreal, Canada
prudencio@valencediscovery.com

## Abstract

In recent years, the Transformer architecture has proven to be very successful in sequence processing, but its application to other data structures, such as graphs, has remained limited due to the difficulty of properly defining positions. Here, we present the *Spectral Attention Network* (SAN), which uses a learned positional encoding (LPE) that can take advantage of the full Laplacian spectrum to learn the position of each node in a given graph. This LPE is then added to the node features of the graph and passed to a fully-connected Transformer. By leveraging the full spectrum of the Laplacian, our model is theoretically powerful in distinguishing graphs, and can better detect similar sub-structures from their resonance. Further, by fully connecting the graph, the Transformer does not suffer from over-squashing, an information bottleneck of most GNNs, and enables better modeling of physical phenomenons such as heat transfer and electric interaction. When tested empirically on a set of 4 standard datasets, our model performs on par or better than state-of-the-art GNNs, and outperforms any attention-based model by a wide margin, becoming the first fully-connected architecture to perform well on graph benchmarks.

## 1 Introduction

The prevailing strategy for graph neural networks (GNNs) has been to directly encode graph structure structure through a sparse message-passing process [17, 19]. In this approach, vector messages are iteratively passed between nodes that are connected in the graph. Multiple instantiations of this message-passing paradigm have been proposed, differing in the architectural details of the message-passing apparatus (see [19] for a review).

However, there is a growing recognition that the message-passing paradigm has inherent limitations. The expressive power of message passing appears inexorably bounded by the Weisfeiler-Lehman isomorphism hierarchy [29, 30, 39]. Message-passing GNNs are known to suffer from pathologies, such

---

*Equal contribution.

35th Conference on Neural Information Processing Systems (NeurIPS 2021).

as *oversmoothing*, due to their repeated aggregation of local information [19], and *over-squashing*, due to the exponential blow-up in computation paths as the model depth increases [1].

As a result, there is a growing interest in deep learning techniques that encode graph structure as a *soft inductive bias*, rather than as a hard-coded aspect of message passing [14, 24]. A central issue with message-passing paradigm is that input graph structure is encoded by restricting the structure of the model's computation graph, inherently limiting its flexibility. This reminds us of how early recurrent neural networks (RNNs) encoded sequential structure via their computation graph—a strategy that leads to well-known pathologies such as the inability to model long-range dependencies [20].

There is a growing trend across deep learning towards more flexible architectures, which avoid strict and structural inductive biases. Most notably, the exceptionally successful Transformer architecture removes any structural inductive bias by encoding the structure via soft inductive biases, such as positional encodings [36]. In the context of GNNs, the self-attention mechanism of a Transformer can be viewed as passing messages between all nodes, regardless of the input graph connectivity.

Prior work has proposed to use attention in GNNs in different ways. First, the GAT model [37] proposed local attention on pairs of nodes that allows a learnable convolutional kernel. The GTN work [42] has improved on the GAT for node and link predictions while keeping a similar architecture, while other message-passing approaches have used enhancing spectral features [8, 13] . More recently, the GT model [14] was proposed as a generalization of Transformers to graphs, where they experimented with sparse and full graph attention while providing low-frequency eigenvectors of the Laplacian as positional encodings.

In this work, we offer a principled investigation of how Transformer architectures can be applied in graph representation learning. **Our primary contribution** is the development of novel and powerful learnable positional encoding methods, which are rooted in spectral graph theory. Our positional encoding technique — and the resulting *spectral attention network (SAN)* architecture — addresses key theoretical limitations in prior graph Transformer work [14] and provably exceeds the expressive power of standard message-passing GNNs. We show that full Transformer-style attention provides consistent empirical gains compared to an equivalent sparse message-passing model, and we demonstrate that our SAN architecture is competitive with or exceeding the state-of-the-art on several well-known graph benchmarks. An overview of the entire method is presented in Figure 1, with a link to the code here: `https://github.com/DevinKreuzer/SAN`.

## 2   Theoretical Motivations

There can be a significant loss in structural information if naively generalizing Transformers to graphs. To preserve this information as well as local connectivity, previous studies [37, 14] have proposed to use the eigenfunctions of their Laplacian as positional encodings. Taking this idea further by using the full expressivity of eigenfunctions as positional encodings, we can propose a principled way of understanding graph structures using their spectra. The advantages of our methods compared to previous studies [37, 14] are shown in Table 1.

Table 1: Comparison of the properties of different graph Transformer models.

| MODELS | GAT | GT sparse | GT full | SAN |
|---|---|---|---|---|
| Preserves local structure in attention | ✓ | ✓ | ✗ | ✓ |
| Uses edge features | ✗ | ✓ | ✗ | ✓ |
| Connects non-neighbouring nodes | ✗ | ✗ | ✓ | ✓ |
| Uses eigenvector-based PE for attention | ✗ | ✓ | ✓ | ✓ |
| Use a PE with structural information | ✗ | ✓ | ✗ | ✓ |
| Considers the ordering of the eigenvalues | ✗ | ✓ | ✓ | ✓ |
| Invariant to the norm of the eigenvector | - | ✓ | ✓ | ✓ |
| Considers the spectrum of eigenvalues | ✗ | ✗ | ✗ | ✓ |
| Considers variable # of eigenvectors | - | ✗ | ✗ | ✓ |
| Aware of eigenvalue multiplicities | - | ✗ | ✗ | ✓ |
| Invariant to the sign of the eigenvectors | - | ✗ | ✗ | ✗ |

---

[1]Presented results add full connectivity before computing the eigenvectors, thus losing the structural information of the graph.

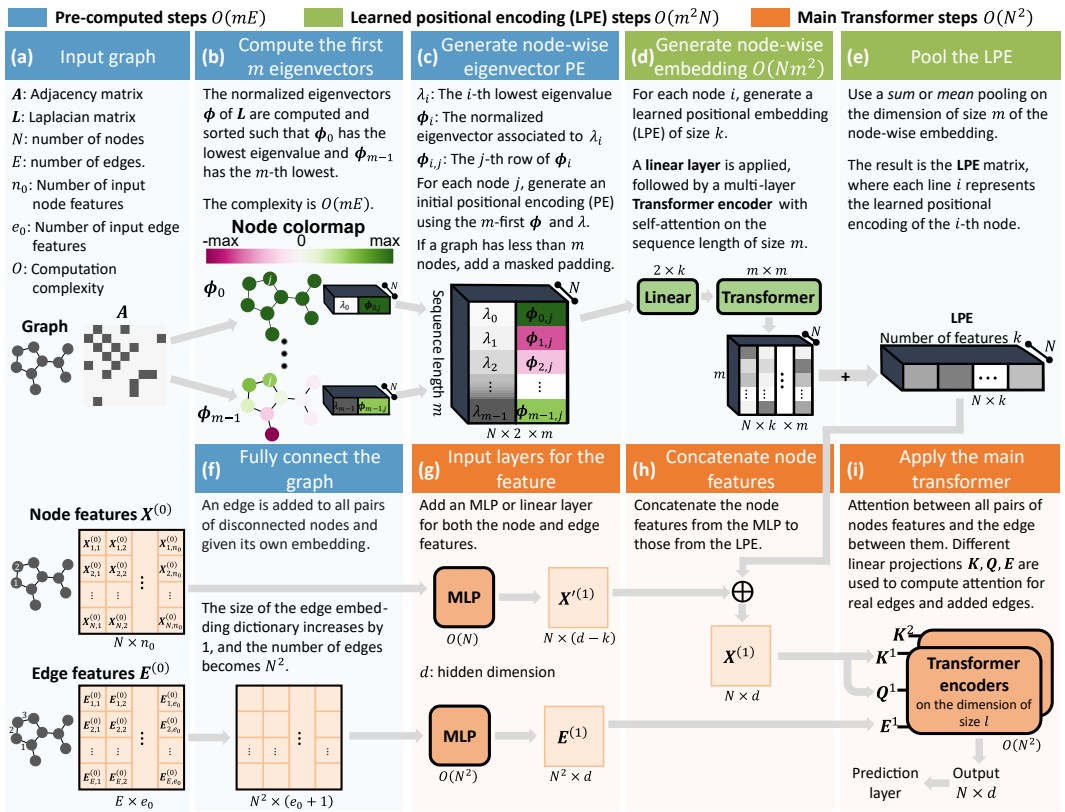

Figure 1: The proposed SAN model with the node LPE, a generalization of Transformers to graphs.

## 2.1 Absolute and relative positional encoding with eigenfunctions

The notion of positional encodings (PEs) in graphs is not a trivial concept, as there exists no canonical way of ordering nodes or defining axes. In this section, we investigate how eigenfunctions of the Laplacian can be used to define absolute and relative PEs in graphs, to measure physical interactions between nodes, and to enable "hearing" of specific sub-structures - similar to how the sound of a drum can reveal its structure.

### 2.1.1 Eigenfunctions equate to sine functions over graphs

In the Transformer architecture, a fundamental aspect is the use of sine and cosine functions as PEs for sequences [36]. However, sinusoids cannot be clearly defined for arbitrary graphs, since there is no clear notion of position along an axis. Instead, their equivalent is given by the eigenvectors $\phi$ of the graph Laplacian $L$. Indeed, in a Euclidean space, the Laplacian (or Laplace) operator corresponds to the divergence of the gradient and its eigenfunctions are sine/cosine functions, with the squared frequencies corresponding to the eigenvalues (we sometimes interchange the two notions from here on). Hence, in the graph domain, the eigenvectors of the graph Laplacian are the natural equivalent of sine functions, and this intuition was employed in multiple recent works which use the eigenvectors as PEs for GNNs [15], for directional flows [4] and for Transformers [14].

Being equivalent to sine functions, we naturally find that the Fourier Transform of a function $\mathscr{F}[f]$ applied to a graph gives $\mathscr{F}[f](\lambda_i) = \langle f, \phi_i \rangle$, where the eigenvalue is considered as a position in the Fourier domain of that graph [6]. Thus, the eigenvectors are best viewed as vectors positioned on the axis of eigenvalues rather than components of a matrix as illustrated in Figure 2.

### 2.1.2 What do eigenfunctions tell us about relative positions?

In addition to being the analog of sine functions, the eigenvectors of the Laplacian also hold important information about the physics of a system and can reveal distance metrics. This is not surprising as

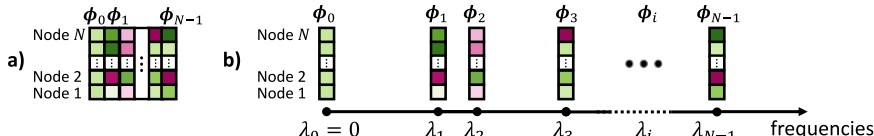

Figure 2: a) Standard view of the eigenvectors as a matrix. b) Eigenvectors $\phi_i$ viewed as vectors positionned on the axis of frequencies (eigenvalues).

the Laplacian is a fundamental operator in physics and is notably used in Maxwell's equations [16] and the heat diffusion [6].

In electromagnetic theory, the (pseudo)inverse of the Laplacian, known in mathematics as the Green's function of the Laplacian [9], represents the electrostatic potential of a given charge. In a graph, the same concept uses the pseudo-inverse of the Laplacian $G$ and can be computed by its eigenfunctions. See equation 1 , where $G(j_1, j_2)$ is the electric potential between nodes $j_1$ and $j_2$, $\hat{\phi}_i$ and $\hat{\lambda}_i$ are the $i$-th eigenvectors and eigenvalues of the symmetric Laplacian $D^{\frac{-1}{2}} L D^{\frac{-1}{2}}$, and $D$ is the degree matrix, and $\hat{\phi}_{i,j}$ the $j$-th row of the vector.

$$G(j_1, j_2) = d_{j_1}^{\frac{1}{2}} d_{j_2}^{\frac{-1}{2}} \sum_{i>0} \frac{(\hat{\phi}_{i,j_1} \hat{\phi}_{i,j_2})^2}{\hat{\lambda}_i} \tag{1}$$

Further, the original solution of the heat equation given by Fourier relied on a sum of sines/cosines known as a Fourier series [7]. As eigenvectors of the Laplacian are the analogue of these functions in graphs, we find similar solutions. Knowing that heat kernels are correlated to random walks [6, 4], we use the interaction between two heat kernels to define in equation 2 the diffusion distance $d_D$ between nodes $j_1, j_2$ [6, 10]. Similarly, the biharmonic distance $d_B$ was proposed as a better measure of distances [28]. Here we use the eigenfunctions of the regular Laplacian $L$.

$$d_D^2(j_1, j_2) = \sum_{k>0} e^{-2t\lambda_i} (\phi_{i,j_1} - \phi_{i,j_2})^2 \quad , \quad d_B^2(j_1, j_2) = \sum_{i>0} \frac{(\phi_{i,j_1} - \phi_{i,j_2})^2}{\lambda_i^2} \tag{2}$$

There are a few things to note from these equations. Firstly, they highlight the importance of pairing *eigenvectors and their corresponding eigenvalues* when supplying information about relative positions in a graph. Secondly, we notice that the product of eigenvectors is proportional to the electrostatic interaction, while the subtraction is proportional to the diffusion and biharmonic distances. Lastly, there is a consistent pattern across all 3 equations: smaller frequencies/eigenvalues are more heavily weighted when determining distances between nodes.

### 2.1.3 Hearing the shape of a graph and its sub-structures

Another well-known property of eigenvalues is how they can be used to discriminate between different graph structures and sub-structures, as they can be interpreted as the frequencies of resonance of the graph. This led to the famous question about whether we can hear the shape of a drum from its eigenvalues [23], with the same questions also applying to geometric objects [12] and 3D molecules [33]. Various success was found with the eigenfunctions being used for partial functional correspondence [32], algorithmic understanding geometries [26], and style correspondence [12]. Examples of eigenvectors for molecular graphs are presented in Figure 3.

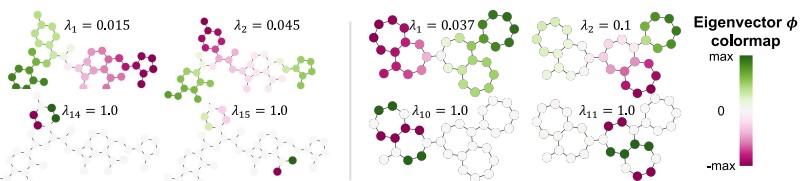

Figure 3: Examples of eigenvalues $\lambda_i$ and eigenvectors $\phi_i$ for molecular graphs. The low-frequency eigenvectors $\phi_1, \phi_2$ are spread accross the graph, while higher frequencies, such as $\phi_{14}, \phi_{15}$ for the left molecule or $\phi_{10}, \phi_{11}$ for the right molecule, often resonate in local structures.

## 2.2 Laplace Eigenfunctions *etiquette*

In Euclidean space and sequences, using sinusoids as PEs is trivial: we can simply select a set of frequencies, compute the sinusoids, and add or concatenate them to the input embeddings, as is done in the original Transformer [36]. However, in arbitrary graphs, reproducing these steps is not as simple since each graph has a unique set of eigenfunctions. In the following section, we present key principles from spectral graph theory to consider when constructing PEs for graphs, most of which have been overlooked by prior methods. They include normalization, the importance of the eigenvalues and their multiplicities, the number of eigenvectors being variable, and sign ambiguities. Our LPE architectures, presented in section 3, aim to address them.

**Normalization**. Given an eigenvalue of the Laplacian, there is an associated eigenspace of dimension greater than 1. To make use of this information in our model, a single eigenvector has to be chosen. In our work, we use the $L_2$ normalization since it is compatible with the definition of the Green's function (1). Thus, we will always chose eigenvectors $\phi$ such that $\langle \phi, \phi \rangle = 1$.

**Eigenvalues**. Another fundamental aspect is that the eigenvalue associated with each eigenvector supplies valuable information. An ordering of the eigenvectors based on their eigenvalue works in sequences since the frequencies are pre-determined. However, this assumption does not work in graphs since the eigenvalues in their spectrum can vary. For example, in Figure 3, we observe how an ordering would miss the fact that both molecules resonate at $\lambda = 1$ in different ways.

**Multiplicities**. Another important problem with choosing eigenfunctions is the possibility of a high multiplicity of the eigenvalues, i.e. when an eigenvalue appears as a root of the characteristic polynomial more than once. In this case, the associated eigenspace may have dimension 2 or more as we can generate a valid eigenvector from any linear combination of eigenvectors with the same eigenvalue. This further complicates the problem of choosing eigenvectors for algorithmic computations and highlights the importance of having a model that can handle this ambiguity.

**Variable number of eigenvectors**. A graph $G_i$ can have at most $N_i$ linearly independent eigenvectors with $N_i$ being its number of nodes. Most importantly, $N_i$ can vary across all $G_i$ in the dataset. Prior work [14] elected to select a fixed number $k$ eigenvectors for each graph, where $k \leq N_i, \forall i$. This produces a major bottleneck when the smallest graphs have significantly fewer nodes than the largest graphs in the dataset since a very small proportion of eigenvectors will be used for large graphs. This inevitably causes loss of information and motivates the need for a model which constructs fixed PEs of dimension $k$, where $k$ does not depend on the number of eigenvectors in the graph.

**Sign invariance**. As noted earlier, there is a sign ambiguity with the eigenvectors. With the sign of $\phi$ being independent of its normalization, we are left with a total of $2^k$ possible combination of signs when choosing $k$ eigenvectors of a graph. Previous work has proposed to do data augmentation by randomly flipping the sign of the eigenvectors [4, 15, 14], and although it can work when $k$ is small, it becomes intractable for large $k$.

## 2.3 Learning with Eigenfunctions

Learning generalizable information from eigenfunctions is fundamental to their succesful usage. Here we detail important points that support it is possible to do so if done correctly.

**Similar graphs have similar spectra**. Thus, we can expect the network to transfer patterns across graphs through the similarity of their spectra. In fact, spectral graph theory tells us that the lowest and largest non-zero eigenvalues are both linked to the geometry of the graph (algebraic connectivity and spectral radius).

**Eigenspaces contain geometric information**. Spectral graph theory has studied the geometric and physical properties of graphs from their Laplacian eigenfunctions in depth. Developing a method that can use the full spectrum of a graph makes it theoretically possible capture this information. It us thus important to capture differences between the full eigenspaces instead of minor differences between specific eigenvalues or eigenvectors from graph to graph.

**Learned positions are relative within graphs**. Eigenspaces are used to understand the relationship between nodes within graphs, not across them. Proposed models should therefore only compare the eigenfunctions of nodes within graphs.

# 3   Model Architecture

In this section, we propose an elegant architecture that can use the eigenfunctions as PEs while addressing the concerns raised in section 2.2. Our *Spectral Attention Network* (SAN) model inputs eigenfunctions of a graph and projects them into a learned positional encoding (LPE) of fixed size. The LPE allows the network to use up to the entire Laplace spectrum of each graph, learn how the frequencies interact, and decide which are most important for the given task.

We propose a two-step learning process summarized earlier in Figure 1. The first step, depicted by blocks (c-d-e) in the figure, applies a Transformer over the eigenfunctions of each node to generate an LPE matrix for each graph. The LPE is then concatenated to the node embeddings (blocks g-h), before being passed to the Graph Transformer (block i). If the task involves graph classification or regression, the final node embeddings are subsequently passed to a final pooling layer.

## 3.1   LPE Transformer Over Nodes

Using Laplace encodings as node features is ubiquitous in the literature concerning the topic. Here, we propose a method for learning node PEs motivated by the principles from section 2.2. The idea of our LPE is inspired by Figure 2, where the eigenvectors $\phi$ are represented as a non-uniform sequence with the eigenvalue $\lambda$ being the position on the frequency axis. With this representation, Transformers are a natural choice for processing them and generating a fixed-size PE.

The proposed LPE architecture is presented in Figure 4. First, we create an embedding matrix of size $2 \times m$ for each node $j$ by concatenating the $m$-lowest eigenvalues with their associated eigenvectors. Here, $m$ is a hyper-parameter for the maximum number of eigenvectors to compute and is analog to the variable-length sequence for a standard Transformer. For graphs where $m > N$, a masked-padding is simply added. Note that to capture the entire spectrum of all graphs, one can simply select $m$ such that it is equal to the maximum number of nodes a graph has in the dataset. A linear layer is then applied on the dimension of size 2 to generate new embeddings of size $k$. A Transformer Encoder then computes self-attention on the sequence of length $m$ and hidden dimension $k$. Finally, a sum pooling reduces the sequence into a fixed $k$-dimensional node embedding.

The LPE model addresses key limitations of previous graph Transformers and is aligned with the first four *etiquettes* presented in section 2.2. By concatenating the eigenvalues with the normalized eigenvector, this model directly addresses the first three *etiquettes*. Namely, it **normalizes** the eigenvectors, pairs eigenvectors with their **eigenvalues** and treats **the number of eigenvectors as a variable**. Furthermore, the model is aware of **multiplicities** and has the potential to linearly combine or ignore some of the repeated eigenvalues.

However, this method still does not address the limitation that the sign of the pre-computed eigenvectors is arbitrary. To combat this issue, we randomly flip the sign of the pre-computed eigenvectors during training as employed by previous work [15, 14], to promote invariance to the sign ambiguity.

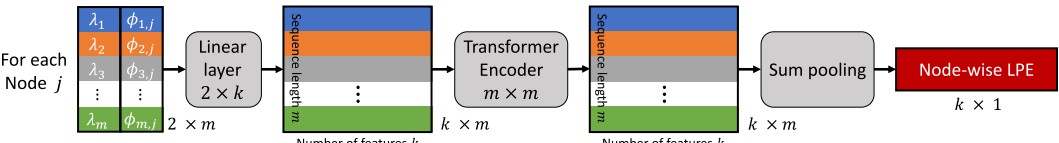

Figure 4: Learned positional encoding (LPE) architectures, with the model being aware of the graph's Laplace spectrum by considering $m$ eigenvalues and eigenvectors, where we permit $m \leq N$, with $N$ denoting the number of nodes. Since the Transformer loops over the nodes, each node can be viewed as an element of a batch to parallelize the computation. Here $\phi_{i,j}$ is the $j$-th element of the eigenvector paired to the $i$-th lowest eigenvalue $\lambda_i$.

## 3.2   LPE Transformer Over Edges

Here we present an alternative formulation for Laplace encodings. This method addresses the same issues as the LPE over nodes, but also resolves the eigenvector sign ambiguity. Instead of encoding *absolute* positions as node features, the idea is to consider *relative* positions encoded as edge features.

Inspired by the physical interactions introduced in 1 and 2, we can take a pair of nodes $(j_1, j_2)$ and obtain **sign-invariant** operators using the absolute subtraction $|\phi_{i,j_1} - \phi_{i,j_2}|$ and the product $\phi_{i,j_1}\phi_{i,j_2}$. These operators acknowledge that the sign of $\phi_{i,j_1}$ at a given node $j_1$ is not important, but that the relative sign between nodes $j_1$ and $j_2$ is important. One might argue that we could directly compute the deterministic values from equations (1, 2) as edge features instead. However, our goal is to construct models that can learn which frequencies to emphasize and are not biased towards the lower frequencies — despite lower frequencies being useful in many tasks.

This approach is only presented thoroughly in appendix A, since it suffers from a major computational bottleneck compared to the LPE over nodes. In fact, for a fully-connected graph, there are $N$ times more edges than nodes, thus the computation complexity is $O(m^2 N^2)$, or $O(N^4)$ considering all eigenfunctions. The same limitation also affects memory and prevents the use of large batch sizes.

## 3.3 Main Graph Transformer

Our attention mechanism in the main Transformer is based on previous work [14], which attempts to repurpose the original Transformer to graphs by considering the graph structure and improving attention estimates with edge feature embeddings.

In the following, note that $\boldsymbol{h}_i^l$ is the $i$-th node's features at the $l$-th layer, and $\boldsymbol{e}_{ij}$ is the edge feature embedding between nodes $i$ and $j$. Our model employs multi-head attention over all nodes:

$$\hat{\boldsymbol{h}}_i^{l+1} = \boldsymbol{O}_h^l \overset{H}{\underset{k=1}{\big\|}} \left(\sum_{j \in V} w_{ij}^{k,l} \boldsymbol{V}^{k,l} \boldsymbol{h}_j^l\right) \tag{3}$$

where $\boldsymbol{O}_h^l \in \mathbb{R}^{d \times d}$, $\boldsymbol{V}^{k,l} \in \mathbb{R}^{d_k \times d}$, $H$ denotes the number of heads, $L$ the number of layers, and $\|$ concatenation. Note that $d$ is the hidden dimension, while $d_k$ is the dimension of a head ($\frac{d}{H} = d_k$).

A key addition from our work is the design of an architecture that performs full-graph attention while preserving local connectivity with edge features via two sets of attention mechanisms: one for nodes connected by real edges in the sparse graph and one for nodes connected by added edges in the fully-connected graph. The attention weights $w_{ij}^{k,l}$ in equation 3 at layer $l$ and head $k$ are given by:

$$\hat{\boldsymbol{w}}_{ij}^{k,l} = \left\{ \begin{array}{ll} \dfrac{\boldsymbol{Q}^{1,k,l}\boldsymbol{h}_i^l \circ \boldsymbol{K}^{1,k,l}\boldsymbol{h}_j^l \circ \boldsymbol{E}^{1,k,l}\boldsymbol{e}_{ij}}{\sqrt{d_k}} & \text{if } i \text{ and } j \text{ are connected in sparse graph} \\[3ex] \dfrac{\boldsymbol{Q}^{2,k,l}\boldsymbol{h}_i^l \circ \boldsymbol{K}^{2,k,l}\boldsymbol{h}_j^l \circ \boldsymbol{E}^{2,k,l}\boldsymbol{e}_{ij}}{\sqrt{d_k}} & \text{otherwise} \end{array} \right\} \tag{4}$$

$$w_{ij}^{k,l} = \left\{ \begin{array}{ll} \dfrac{1}{1+\gamma} \cdot \text{softmax}(\sum_{d_k} \hat{\boldsymbol{w}}_{ij}^{k,l}) & \text{if } i \text{ and } j \text{ are connected in sparse graph} \\[3ex] \dfrac{\gamma}{1+\gamma} \cdot \text{softmax}(\sum_{d_k} \hat{\boldsymbol{w}}_{ij}^{k,l}) & \text{otherwise} \end{array} \right\} \tag{5}$$

where $\circ$ denotes element-wise multiplication and $\boldsymbol{Q}^{1,k,l}, \boldsymbol{Q}^{2,k,l}, \boldsymbol{K}^{1,k,l}, \boldsymbol{K}^{2,k,l}, \boldsymbol{E}^{1,k,l}, \boldsymbol{E}^{2,k,l} \in \mathbb{R}^{d_k \times d}$. $\gamma \in \mathbb{R}^+$ is a hyperparameter which tunes the amount of bias towards full-graph attention, allowing flexibility of the model to different datasets and tasks where the necessity to capture long-range dependencies may vary. Note that softmax outputs are clamped between $-5$ and $5$ for numerical stability and that the keys, queries and edge projections are different for pairs of connected nodes $(\boldsymbol{Q}^1, \boldsymbol{K}^1, \boldsymbol{E}^1)$ and disconnected nodes $(\boldsymbol{Q}^2, \boldsymbol{K}^2, \boldsymbol{E}^2)$.

A multi-layer perceptron (MLP) with residual connections and normalization layers are then applied to update representations, in the same fashion as the GT method [14].

$$\hat{\hat{\boldsymbol{h}}}^{l+1} = \text{Norm}(\boldsymbol{h}_i^l + \hat{\boldsymbol{h}}_i^{l+1}), \quad \hat{\hat{\boldsymbol{h}}}_i^{l+1} = \boldsymbol{W}_2^l \text{ReLU}(\boldsymbol{W}_1^l \hat{\boldsymbol{h}}_i^l), \quad \boldsymbol{h}_i^{l+1} = \text{Norm}(\hat{\hat{\boldsymbol{h}}}^{l+1} + \hat{\hat{\boldsymbol{h}}}_i^{l+1}) \tag{6}$$

with the weight matrices $\boldsymbol{W}_1^l \in \mathbb{R}^{2d \times d}$, $\boldsymbol{W}_2^l \in \mathbb{R}^{d \times 2d}$. Edge representations are not updated as it adds complexity with little to no performance gain. Bias terms are omitted for presentation.

## 3.4 Limitations

The first limitation of the node-wise LPE, and noted in Table 1 is the lack of sign invariance of the model. A random sign-flip of an eigenvector can produce different outputs for the LPE, meaning

that the model needs to learn a representation invariant to these flips. We resolve this issue with the edge-wise LPE proposed in 3.2, but it comes at a computational cost.

Another limitation of the approach is the computational complexity of the LPE being $O(m^2 N)$, or $O(N^3)$ if considering all eigenfunctions. Further, as nodes are batched in the LPE, the total memory on the GPU will be *num_params * num_nodes_in_batch* instead of *num_params * batch_size*. Although this is limiting, the LPE is not parameter hungry, with $k$ usually kept around 16. Most of the model's parameters are in the *Main Graph Transformer* of complexity $O(N^2)$.

Despite Transformers having increased complexity, they managed to revolutionize the NLP community. We argue that to shift away from the message-passing paradigm and generalize Transformers to graphs, it is natural to expect higher computational complexities. This is exacerbated by sequences being much simpler to understand than graphs due to their linear structure. Future work could overcome this by using variations of Transformers that scale linearly or logarithmically [34].

## 3.5 Theoretical properties of the architecture

Due to the full connectivity, it is trivial that our model does not suffer from the same limitations in expressivity as its convolutional/message-passing counterpart.

**WL test and universality**. The DGN paper [4] showed that using the eigenvector $\phi_1$ is enough to distinguish some non-isomorphic graphs indistinguishable by the 1-WL test.

Given that our model uses the full set of eigenfunctions, and given enough parameters, our model can distinguish any pair of non-isomorphic graphs and is more powerful than any WL test in that regard. However, this does not solve the graph isomorphism problem in polynomial time; it only approximates a solution, and the number of parameters required is unknown and possibly non-polynomial. In appendix C, we present a proof of our statement, and discuss why the WL test is not well suited to study the expressivity of graph Transformers due to their universality.

**Reduced over-squashing**. Over-squashing represents the difficulty of a graph neural network to pass information to distant neighbours due to the exponential blow-up in computational paths [1].

For the fully-connected network, it is trivial to see that over-squashing is non-existent since there are direct paths between distant nodes.

**Physical interactions**. Another point to consider is the ability of the network to learn physical interactions between nodes. This is especially important when the graph models physical, chemical, or biological structures, but can also help understanding pixel interaction in images [2, 3]. Here, we argue that our SAN model, which uses the Laplace spectrum more effectively, can learn to mimic the physical interactions presented in section 2.1.2. This contrasts with the convolutional approach that requires deep layers for the receptive field to capture long-distance interactions. It also contrasts with the GT model [14], which does not use eigenvalues or enough eigenfunctions to properly model physical interactions in early layers. However, due to the lack of sign-invariance in the proposed node-wise LPE, it is difficult to learn these interactions accurately. The edge-wise LPE (section 3.2) could be better suited for the problem, but it suffers from higher computational complexity.

## 4 Experimental Results

The model is implemented in PyTorch [31] and DGL [38] and tested on established benchmarks from [15] and [21] provided under MIT license. Specifically, we applied our method on ZINC, PATTERN, CLUSTER, MolHIV and MolPCBA, while following their respective training protocols with minor changes, as detailed in the appendix B.1. The computation time and hardware is provided in appendix B.4.

We first conducted an ablation study to fairly compare the benefits of using full attention and/or the node LPE. We then took the best-performing model, tuned some of its hyperparameters, and matched it up against the current state-of-the-art methods. Since we use a similar attention mechanism, our code was developed on top of the code from the GT paper [14], provided under the MIT license.

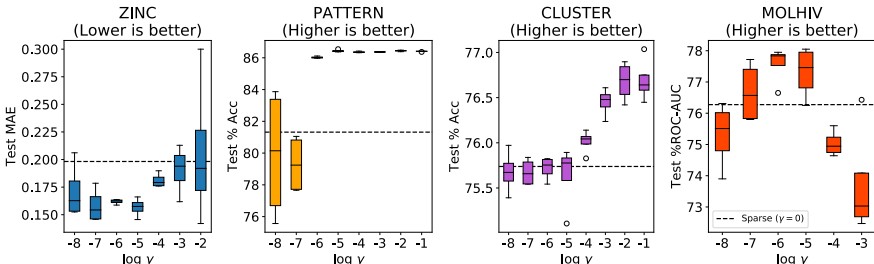

Figure 5: Effect of the $\gamma$ parameter on the performance across datasets from [15, 21], using the Node LPE. Dotted black lines indicate sparse attention, which is equivalent to setting $\gamma = 0$. Each box plot consists of 4 runs, with different seeds (except MolHIV).

| Model details | | ZINC | PATTERN | CLUSTER | MOLHIV | | |
|---|---|---|---|---|---|---|---|
| **Attention** | **LPE** | **MAE** | **% ACC** | **% ACC** | **% ROC-AUC** | | |
| Sparse | - | $0.267 \pm 0.032$ | $83.613 \pm 0.663$ | $75.683 \pm 0.098$ | $73.46 \pm 0.71$ | | Best |
| Sparse | Node | $0.198 \pm 0.004$ | $81.329 \pm 2.150$ | $75.738 \pm 0.106$ | $76.61 \pm 0.62$ | | |
| Full | - | $0.392 \pm 0.055$ | $86.322 \pm 0.049$ | $76.447 \pm 0.177$ | $73.84 \pm 1.80$ | | |
| Full | Node | $0.157 \pm 0.006$ | $86.441 \pm 0.040$ | $76.691 \pm 0.247$ | $77.57 \pm 0.61$ | | Worst |

Figure 6: Ablation study on datasets from [15, 21] for the node LPE and full graph attention, with no hyperparameter tuning other than $\gamma$ taken from Figure 5. For a given dataset, all models use the same hyperparameters, but the hidden dimensions are adjusted to have $\sim 500k$ learnable parameters. Means and uncertainties are derived from four runs, with different seeds (except MolHIV).

## 4.1 Sparse vs. Full Attention

To study the effect of incorporating full attention, we present an ablation study of the $\gamma$ parameter in Figure 5. We remind readers that $\gamma$ is used in equation 5 to balance between *sparse* and *full* attention. Setting $\gamma = 0$ strictly enables sparse attention, while $\gamma = 1$ does not bias the model in any direction.

It is apparent that molecular datasets, namely ZINC and MOLHIV, benefit less from full attention, with the best parameter being $\log \gamma \in (-7, -5)$. On the other hand, the larger SBM datasets (PATTERN and CLUSTER) benefit from a higher $\gamma$ value. This can be explained by the fact that molecular graphs rely more on understanding local structures such as the presence of rings and specific bonds, especially in the artificial task from ZINC which relies on counting these specific patterns [15]. Furthermore, molecules are generally smaller than SBMs. As a result, we would expect less need for full attention, as information between distant nodes can be propagated with few iterations of even sparse attention. We also expect molecules to have fewer multiplicities, thus reducing the space of eigenvectors. Lastly, the performance gains in using full attention on the CLUSTER dataset can be attributed to it being a semi-supervised task, where some nodes within each graph are assigned their true labels. With full attention, every node receives information from the labeled nodes at each iteration, reinforcing confidence about the community they belong to.

In Figure 6, we present another ablation study to measure the impact of the node LPE in both the *sparse* and *full* architectures. We observe that the proposed node-wise LPE contributes significantly to the performance for molecular tasks (ZINC and MOLHIV), and believe that it can be attributed to the detection of substructures (see Figure 3). For PATTERN and CLUSTER, the improvement is modest as the tasks are simple clustering [15]. Previous work even found that the optimal number of eigenvectors to construct PE for PATTERN is only 2 [14].

## 4.2 Comparison to the state-of-the-art

When comparing to the state-of-the-art (SOTA) models in the literature in Figure 7, we observe that our SAN model consistently performs better on all synthetic datasets from [15], highlighting the strong expressive power of the model. On the MolHIV dataset, the performance on the test set is slightly lower than the SOTA. However, the model performs better on the validation set (85.30%) in comparison to PNA (84.25%) and DGN (84.70%). This can be attributed to a well-known issue with this dataset: the validation and test metrics have low correlation. In our experiments, we found higher test results with lower validation scores when restricting the number of epochs. Here, we also included results on the MolPCBA dataset, where we witnessed competitive results as well.

Other top-performing models, namely PNA [11] and DGN [4], use a message-passing approach [17] with multiple aggregators. When compared to attention-based models, SAN consistently outperforms the SOTA by a wide margin. To the best of our knowledge, SAN is the first fully-connected model to perform well on graph tasks, as is evident by the poor performance of the *GT (full)* model.

| | ZINC | PATTERN | CLUSTER | MOLHIV | MOLPCBA |
|---|---|---|---|---|---|
| Model | MAE | % Acc | % Acc | % ROC-AUC | % AP |
| GCN | $0.367 \pm 0.011$ | $71.892 \pm 0.334$ | $68.498 \pm 0.976$ | $76.06 \pm 0.97$ | $20.20 \pm 0.24$ |
| GraphSage | $0.398 \pm 0.002$ | $50.492 \pm 0.001$ | $63.844 \pm 0.110$ | - | - |
| GatedGCN | $0.282 \pm 0.015$ | $85.568 \pm 0.088$ | $73.840 \pm 0.326$ | - | - |
| GatedGCN-PE | $0.214 \pm 0.013$ | $86.508 \pm 0.085$ | $76.082 \pm 0.196$ | | |
| GIN | $0.526 \pm 0.013$ | $85.387 \pm 0.136$ | $64.716 \pm 1.553$ | $75.58 \pm 1.40$ | $22.66 \pm 0.28$ |
| PNA | $0.142 \pm 0.010$ | - | - | $79.05 \pm 1.32$ | $28.38 \pm 0.35$ |
| DGN | - | - | - | $\mathbf{79.70 \pm 0.97}$ | $\mathbf{28.85 \pm 0.30}$ |
| Attention-based | | | | | |
| GAT | $0.384 \pm 0.007$ | $78.271 \pm 0.186$ | $70.587 \pm 0.447$ | - | - |
| GT (sparse) | $0.226 \pm 0.014$ | $84.808 \pm 0.068$ | $73.169 \pm 0.662$ | - | - |
| GT (full) | $0.598 \pm 0.049$ | $56.482 \pm 3.549$ | $27.121 \pm 8.471$ | - | - |
| SAN | $\mathbf{0.139 \pm 0.006}$ | $\mathbf{86.581 \pm 0.037}$ | $\mathbf{76.691 \pm 0.65}$ | $77.85 \pm 0.247$ | $27.65 \pm 0.42$ |

Figure 7: Comparing our tuned model on datasets from [15, 21], against GCN [25], GraphSage [18], GIN [39], GAT [37], GatedGCN [5], PNA [11], and DGN [4]. Means and uncertainties are derived from four runs with different seeds, except MolHIV which uses 10 runs with identical seed. The number of parameters is fixed to $\sim 500k$ for ZINC, PATTERN and CLUSTER.

## 5 Conclusion

In summary, we presented the SAN model for graph neural networks, a new Transformer-based architecture that is aware of the Laplace spectrum of a given graph from the learned positional encodings. The model was shown to perform on par or better than the SOTA on multiple benchmarks and outperforms other Attention-based models by a large margin. As is often the case with Transformers, the current model suffers from a computational bottleneck, and we leave it for future work to implement variations of Transformers that scale linearly or logarithmically. This will enable the edge-wise LPE presented in appendix A, a theoretically more powerful version of the SAN model.

**Societal Impact**. The presented work is focused on theoretical and methodological improvements to graph neural networks, so there are limited direct societal impacts. However, indirect negative impacts could be caused by malicious applications developed using the algorithm. One such example is the tracking of people on social media by representing their interaction as graphs, thus predicting and influencing their behavior towards an external goal. It also has an environmental impact due to the greater energy use that arises from the computational cost $O(m^2N + N^2)$ being larger than standard message passing or convolutional approaches of $O(E)$.

**Funding Disclosure**.

Devin Kreuzer is supported by an NSERC grant.

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
