# A   LPE Transformer Over Edges

Consider one of the most fundamental notions in physics; *Potential energy*. Interestingly, potential energy is always measured as a potential difference; it is not an inherent individual property, such as mass. Strikingly, it is also the *relative* Laplace embeddings of two nodes that paint the picture, as a node's Laplace embedding on its own reveals no information at all. With this in mind, we argue that Laplace positional encodings are more naturally represented as edge features, which encode a notion of *relative* position of the two endpoints in the graph. This can be viewed as a distance encoding, which was shown to improve the performance of node and link prediction in GNNs [27].

The formulation is very similar to the method for learning positional node embeddings. Here, a Transformer Encoder is applied on each graph by treating edges as a batch of variable size and eigenvectors as a variable sequence length. We again compute up to the $m$-lowest eigenvectors with their eigenvalues but, instead of directly using the eigenvector elements, we compute the following vectors:

$$|\phi_{:,j_1} - \phi_{:,j_2}| \qquad (7) \qquad\qquad \phi_{:,j_1} \circ \phi_{:,j_2} \qquad (8)$$

where ":" denotes along all up to $m$ eigenvectors, and $\circ$ denotes element-wise multiplication. Note that these new vectors are completely invariant to sign permutations of the precomputed eigenvectors.

As per the LPE over nodes, the 3-length vectors are expanded with a linear layer to generate embeddings of size $k$ before being input to the Transformer Encoder. The final embeddings are then passed to a sum pooling layer to generate fixed-size edge positional encodings, which are then used to compute attention weights in equation 4.

This method addresses **all etiquettes** raised in section 2.2. However, it suffers from a major computational bottleneck compared to the LPE over nodes. Indeed, for a fully-connected graph, there are $N$ times more edges than nodes, thus the computation complexity is $O(m^2N^2)$, or $O(N^4)$ considering all eigenfunctions. This same limitation also affects the memory, as efficiently batching the $N^2$ edges will increase the memory consumption of the LPE by a drastic amount, preventing the model from using large batch sizes and making it difficult to train.

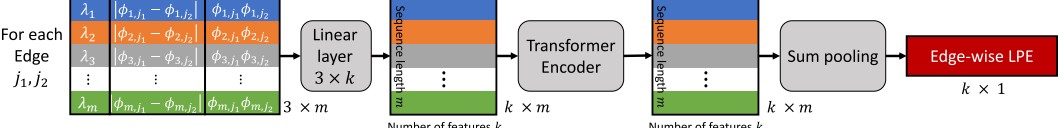

Figure 8: Edge-wise Learned positional encoding (LPE) architectures, where the relative position is considered instead of the absolute position. The model is aware of the graph's Laplace spectrum by considering $m$ eigenvalues and eigenvectors, where we permit $m \leq N$, with $N$ denoting the number of nodes. Since the Transformer loops over the edges, each edge can be viewed as an element of a batch to parallelize the computation. The computational complexity is $O(m^2E)$ or $O(m^2N^2)$ for a fully-connected graph.

# B   Appendix - Implementation details

## B.1   Benchmarks and datasets

To test our models' performance, we rely on standard benchmarks proposed by [15] and [21] and provided under the MIT license. In particular, we chose ZINC, PATTERN, CLUSTER, and MolHIV.

**ZINC [15]**. A synthetic molecular graph regression dataset, where the predicted score is given by the subtraction of computationally estimated properties $logP - SA$. Here, $logP$ is the computed octanol-water partition coefficient, and $SA$ is the synthetic accessibility score [22].

**CLUSTER [15]**. A synthetic benchmark for node classification. The graphs are generated with Stochastic Block Models, a type of graph used to model communities in social networks. In total, 6 communities are generated and each community has a single node with its true label assigned. The task is to classify which nodes belong to the same community.

**PATTERN [15]**. A synthetic benchmark for node classification. The graphs are generated with Stochastic Block Models, a type of graph used to model communities in social networks. The task is to classify the nodes into 2 communities, testing the GNNs ability to recognize predetermined subgraphs.

**MolHIV [21]**. A real-world molecular graph classification benchmark. The task is to predict whether a molecule inhibits HIV replication or not. The molecules in the training, validation, and test sets are divided using a scaffold splitting procedure that splits the molecules based on their two-dimensional structural frameworks. The dataset is heavily imbalanced towards negative samples. It is also known that this dataset suffers from a strong de-correlation between validation and test set performance, meaning that more hyperparameter fine-tuning on the validation set often leads to lower test set results.

**MolPCBA [21]**. Another real-world molecular graph classification benchmark. The dataset is larger than MolHIV and applies a similar scaffold spliting procedure. It consists of multiple, extremely skewed (only 1.4% positivity) molecular classification tasks, and employs Average Precision (AP) over them as a metric.

## B.2 Ablation studies

The results in Figures 5-6 are done as an ablation study with a minimal tuning of the hyperparameters of the network to measure the impact of the node LPE and full attention. A majority of the hyperparameters used were tuned in previous work [15]. However, we altered some of the existing parameters to accommodate the parameter-heavy LPE, and modified the Main Graph Transformer hidden dimension such that all models have approximately $\sim 500k$ parameters for a fair comparison. We present results for the full attention with the optimal $\gamma$ value optimized on the Node LPE model. We did this to isolate the impact that the Node LPE has on improving full attention. Details concerning the model architecture parameters are visible in Figure 9.

| Attention | LPE | LPE layers | LPE dimension | GT layers | GT hidden dimension | #Parameters |
|---|---|---|---|---|---|---|
| **ZINC** | | | | | | |
| †Sparse | - | - | - | 6 | 96 | 511201 |
| Sparse | Node | 3 | 16 | 6 | 72 | 494865 |
| Full | - | - | - | 6 | 80 | 471361 |
| Full | Node | 3 | 16 | 6 | 64 | 508577 |
| **PATTERN** | | | | | | |
| Sparse | - | - | - | 6 | 96 | 508634 |
| Sparse | Node | 3 | 16 | 6 | 72 | 493340 |
| Full | - | - | - | 6 | 80 | 469142 |
| Full | Node | 3 | 16 | 6 | 64 | 507202 |
| **CLUSTER** | | | | | | |
| Sparse | - | - | - | 16 | 56 | 461348 |
| Sparse | Node | 1 | 16 | 16 | 56 | 530036 |
| Full | - | - | - | 16 | 48 | 450498 |
| Full | Node | 1 | 16 | 16 | 48 | 519186 |
| **MOLHIV** | | | | | | |
| Sparse | - | - | - | 6 | 96 | 525985 |
| Sparse | Node | 2 | 16 | 6 | 80 | 503265 |
| Full | - | - | - | 6 | 80 | 483601 |
| †Full | Node | 2 | 16 | 6 | 72 | 528265 |

Figure 9: Model architecture parameters for the ablation study. We modify the hidden dimensions of the Main Graph Transformer (GT) such that all models have $\sim 500k$ parameters for a fair comparison. †The batch size was doubled to ensure convergence of the model. All other parameters outside the GT hidden dimension are consistent within a dataset experiment.

For the training parameters, we employed an Adam optimizer with a learning rate decay strategy initialized in $\{10^{-3}, 10^{-4}\}$ as per [15], with some minor modifications:

**ZINC [15]**. We selected an *initial learning rate* of $7 \times 10^{-4}$ and increased the *patience* from 10 to 25 to ensure convergence. **PATTERN [15]**. We selected an *initial learning rate* of $5 \times 10^{-4}$. **CLUSTER [15]**. We selected an *initial learning rate* of $5 \times 10^{-4}$ and reduced the *minimum learning rate* from $10^{-6}$ to $10^{-5}$ to speed up training time. **MolHIV [21]**. We elected to use similar training

procedures for consistency. We selected an *initial learning rate* of $10^{-4}$, a *reduce factor* of $0.5$, a *patience* of $20$, a *minimum learning rate* of $10^{-5}$, a *weight decay* of $0$ and a *dropout* of $0.03$.

## B.3 SOTA Comparison study

For the results in Figure 7, we tuned some of the hyperparameters, using the following strategies. The optimal parameters are in **bold**.

**ZINC**. Due to the $500k$ parameter budget, we tuned the pairing $\{GT\ layers,\ GT\ hidden\ dimension\} \in \{\{6, 72\}, \{8, 64\}, \mathbf{\{10, 56\}}\}$ and *readout* $\in \{$"mean", **"sum"**$\}$ **PATTERN**. Due to the $500k$ parameter budget and long training times, we only tuned the pairing $\{GT\ layers,\ GT\ hidden\ dimension\} \in \{\mathbf{\{4, 80\}}, \{6, 64\}\}$ **CLUSTER**. Due to the $500k$ parameter budget and long training times, we only tuned the pairing $\{GT\ layers,\ GT\ hidden\ dimension\} \in \{\{12, 64\}, \mathbf{\{16, 48\}}\}$ **MolHIV**. With no parameter budget, we elected to do a more extensive parameter tuning in a two-step process while measuring validation metrics on 3 runs with identical seeds.

1. We tuned *LPE dimension* $\in \{8, \mathbf{16}\}$, *GT layers* $\in \{4, 6, 8, \mathbf{10}\}$, *GT hidden dimension* $\in \{48, \mathbf{64}, 72, 80, 96\}$

2. With the highest performing validation model from step 1, we then tuned *dropout* $\in \{0, \mathbf{0.01}, 0.025\}$ and *weight decay* $\in \{\mathbf{0}, 10^{-6}, 10^{-5}\}$

With the final optimized parameters, we reran 10 experiments with identical seeds.

**MolPCBA**. With no parameter budget, we elected to do a more extensive parameter tuning as well. We tuned *learning rate* $\in \{0.0001, \mathbf{0.0003}, 0.0005\}$, *dropout* $\in \{0, 0.1, 0.2, 0.3, 0.4, \mathbf{0.5}\}$, *GT layers* $\in \{2, 4, \mathbf{5}, 6, 8, 10, 12\}$, *GT layers* $\in \{128, 256, \mathbf{304}, 512\}$, *LPE layers* $\in \{8, \mathbf{10}, 12\}$ amd *LPE dimension* $\in \{8, \mathbf{16}\}$

## B.4 Computation details

| Dataset | Resource | Cluster | GPU | Epoch/Total time |
|---------|----------|---------|-----|------------------|
| ZINC | Compute Canada | Graham | Tesla P100-PCIE (12 GB) | 106s/17.88hrs |
| PATTERN | Compute Canada | Graham | Tesla P100-PCIE (12 GB) | 340s/12.52hrs |
| CLUSTER | Compute Canada | Beluga | Tesla V100-SXM2 (16 GB) | 433s/11.30hrs |
| MOLHIV | Compute Canada | Cedar | Tesla V100-SXM2 (32 GB) | 204s/5.34hrs |
| MOLPCBA | Compute Canada | Cedar | Tesla V100-SXM2 (32 GB) | 883s/48.02hrs |

Figure 10: Computational details for SOTA Comparison study.

## C Expressivity and complexity analysis of graph Transformers

In this section, we discuss how the universality of Transformers translates to graphs when using different node identifiers. Theoretically, this means that by simply labeling each node, Transformers can learn to distinguish any graph, and the WL test is no longer suited to study their expressivity.

Thus, we introduce the notion of learning complexity to better compare each architecture's ability to understand the space of isomorphic graphs. We apply the complexity analysis to the LPE and show that it can more easily capture the structure of graphs than a naive Transformer.

### C.1 Universality of Transformers for sequence-to-sequence approximations

In recent work [40] [41], it was proven that Transformers are universal sequence-to-sequence approximators, meaning that they can encode any function that approximately maps any first sequence into a second sequence when given enough parameters. More formally, they proved the following theorems for the universality of Transformers:

**Theorem 1.** *For any* $1 \leq p < \infty$, $\varepsilon > 0$ *and any function* $f : \mathbf{R}^{d \times n} \to \mathbf{R}^{d \times n}$ *that is equivariant to permutations of the columns, there is a Transformer* $g$ *such that the* $L^p$ *distance between* $f$ *and* $g$ *is smaller than* $\varepsilon$.

Let $\boldsymbol{B}^n$ be the $n$-dimensional closed ball and denote by $C^0(\boldsymbol{B}^{d\times n}, \boldsymbol{R}^{d\times n})$ the set of all continuous functions of the ball to $\boldsymbol{R}^{d\times n}$. A Transformer with positional encoding $g_p$ is a Transformer $g$ such that to each input $\boldsymbol{X}$, a fixed learned positional encoding $\boldsymbol{E}$ is added such that $g_p(\boldsymbol{X}) = g(\boldsymbol{X} + \boldsymbol{E})$.

**Theorem 2.** *For any $1 \leq p < \infty$, $\varepsilon > 0$ and any function $f \in C^0(\boldsymbol{B}^{d\times n}, \boldsymbol{R}^{d\times n})$, there is a Transformer with positional encoding $g$ such that the $L^p$ distance between $f$ and $g$ is smaller than $\varepsilon$.*

### C.2 Graph Transformers approximate solutions to the graph isomorphism problem

We now explore the consequences of the previous 2 theorems on the use of Transformers for graph representation learning. We first describe 2 types of Transformers on graphs; one for node and one for edge inputs. They will be used to deduce corollaries of theorems 1 and 2 for graph learning and later comparison with our proposed architecture. Assume now that all nodes of the graphs we consider are given an integer label in $\{1, ..., N\}$.

The naive edge transformer takes as input a graph represented as a sequence of ordered pairs $((i, j), \sigma_{i,j})$ with $i \leq j$ the indices of 2 vertices and $\sigma_{i,j}$ equal to 1 or 0 if the vertices $i, j$ are connected or not. Recall there are $N(N-1)/2$ pairs of integers $i, j$ in $\{1, ..., N\}$ with $i < j$ the indices of 2 vertices and $\sigma_{i,j}$ equal to 1 or 0 if the vertices $i, j$ are connected or not. It is obvious that any ordering of these edge vectors describe the same graph. Recall there are $N(N-1)/2$ pairs of integers $i, j$ in $\{1, ..., N\}$ with $i \leq j$. Consider the set of functions $f : \boldsymbol{R}^{N(N-1)/2\times 2} \to \boldsymbol{R}^{N(N-1)/2\times 2}$ that are equivariant to the permutations of columns then theorem 1 says the function $f$ can be approximated with arbitrary accuracy by Transformers on edge input.

The naive node Transformer can be defined as a Transformer with positional encodings. This graph Transformer will take as input the identity matrix and as positional encodings the padded adjacency matrix. This can be viewed as a one-hot encoding of each node's neighbors. Consider the set of continuous functions $f : \boldsymbol{R}^{N\times N} \to \boldsymbol{R}^{N\times N}$, then theorem 2 says the function $f$ can be approximated with arbitrary accuracy by Transformers on node inputs.

From these two observations on the universality of graph Transformers, we get as a corollary that these 2 types of Transformers can approximate solutions of the graph isomorphism problem. In each case, pick a function that is invariant under node index permutations and maps non-isomorphic graphs to different values and apply theorem 1 or 2 that shows there is a Transformer approximating that function to an arbitrarily small error in the $L^p$ distance. This is an interesting fact since it is known that the discrimination power of most message passing graph networks is upper bounded by the Weisfeiler-Lehman test which is unable to distinguish some graphs.

This may seem strange since it is unlikely there is an algorithm solving the graph isomorphism problem in polynomial time to the number of nodes $N$, and we address this issue in the notes below.

**Note 1: Only an approximate solution**. The universality theorems do not state that Transformers solve the isomorphism problem, but that they can approximate a solution. They only learn the invariant functions only up to some error so they still can mislabel graphs.

**Note 2: Estimate of number of Transformer blocks**. For the approximation of the function $f$ by a Transformer to be precise, a large number of Transformer blocks will be needed. In [40], it is stated that the universal class of function is obtained by composing Transformer blocks with 2 heads of size 1 followed by a feed-forward layer with 4 hidden nodes. In [41] section 4.1, an estimate of the number of blocks is given. If $f : \boldsymbol{R}^{d\times n} \to \boldsymbol{R}^{d\times n}$ is $L$-Lipschitz, then $||f(\boldsymbol{X}) - f(\boldsymbol{Y})|| < \varepsilon/2$ when $||\boldsymbol{X} - \boldsymbol{Y}|| < \varepsilon/2L = \delta$. In the notation of [41], the LPE has constants $p = 2$ and $s = 1$. If $g$ is a composition of Transformer blocks then an error $||f - g||_{L^p} < \varepsilon$ can be achieved with a number of Transformer blocks larger than

$$\left(\frac{dn}{\delta}\right) + \left(\frac{p(n-1)}{\delta^d} + s\right) + \left(\frac{n}{\delta^{dn}}\right) = \frac{dn2L}{\varepsilon} + \frac{2(n-1)(2L)^d}{\varepsilon^d} + 1 + \frac{n(2L)^{dn}}{\varepsilon^{dn}}$$

In the case of the node encoder described above, $n = d = N$ (the number of nodes) and the last term in the sum above becomes $N(2L/\varepsilon)^{N^2}$, so the number of parameters and therefore the computational time is exponential in the number of nodes for a fixed error $\varepsilon$. Note that this bound on the number of Transformer blocks might not be tight and might be much lower for a specific problem.

**Note 3: Learning invariance to label permutations**. In the above proof, the Transformer is assumed to be able to label all isomorphic graphs into the same class within a small error. However, given a graph of $N$ nodes, there are $N!$ different node labeling permutations, and they all need to be mapped to the same output class. It seems unlikely that such function can be learned with polynomial complexity to $N$.

Following these observations, it does not seem appropriate to compare Transformers to the WL test as is the custom for graph neural networks and we think at this point we should seek a new measure of expressiveness of graph Transformers.

## C.3 Expressivity of the node-LPE

Here, we want to show that the proposed node-LPE can generate a unique node identifier that allows our Transformer model to be a universal approximator on graphs, thus allowing us to approximate a solution to graph isomorphism.

Recall the node LPE takes as input an $N \times m \times 2$ tensor with $m$ the number of eigenvalues and eigenvectors that are used to represent the nodes. The output is a $N \times k$ tensor. Notice that 2 non-isomorphic graphs on $N$ nodes can have the same $m < N$ eigenvalues and eigenspaces and disagree on the last $N - m$ eigenvalues and eigenspaces. Any learning algorithm missing the last $N - m$ pieces of information won't be able to distinguish these graphs. Here we will fix some $m$ and show that the resulting Transformer can approximately classify all graphs with $N \leq m$.

Fix some linear injection $\boldsymbol{M} : \boldsymbol{R}^{N \times 2 \times m} \rightarrow \boldsymbol{R}^{N \times k \times m}$. Let $G$ be a graph and $U \subset \boldsymbol{R}^{N \times 2 \times m}$ be bounded set containing all the tensor representations of graphs $\boldsymbol{T}_G$ and let $R$ be the radius of a ball containing $\boldsymbol{M}(U)$. Consider the set $C^0(\boldsymbol{B}_R^{N \times k \times m}, \boldsymbol{R}^{N \times k \times m})$ of continuous functions of the closed radius $R$ ball in $\boldsymbol{R}^{N \times k \times m}$. Finally, denote by $\boldsymbol{S} : \boldsymbol{R}^{N \times k \times m} \rightarrow \boldsymbol{R}^{N \times k}$ the linear function taking the sum of all values in the $m$ dimension. The following universality result for LPE Transformers is a direct consequence of theorem 2.

**Proposition 1.** *For any $1 \leq p < \infty$, $\varepsilon > 0$ and any continuous function $F : \boldsymbol{B}_R^{N \times k \times m} \rightarrow \boldsymbol{R}^{N \times k}$, there is an LPE Transformer $g$ such that the $L^p$ distance between $M \circ f \circ S$ and $g$ is smaller than $\varepsilon$.*

As a corollary, we get the same kind of approximation to solutions of the graph isomorphism problem as with the naive Transformers. Let $f$ be a function of $C^0(\boldsymbol{B}_R^{N \times k \times m}, \boldsymbol{R}^{N \times k \times m})$ that maps $M(\boldsymbol{T}_G)$ to a value that is only dependent of the isomorphism class of the graph and assigns different values to different isomorphism classes. We can further assume that $f$ takes values that are 0 for all but one coordinate in the $k$ dimension. The same type of argument is possible for the edge-LPE from figure 8.

## C.4 Comparison of the learning complexity of naive graph Transformers and LPE

We now argue that while the LPE Transformer and the naive graph Transformers of section C.2 can all approximate a function $f$ solution of the graph isomorphism problem, the complexity of the learning problem of the LPE is much lower since the spaces it has to learn are simpler.

**Naive Node Transformer**. First recall that the naive node Transformer learns a map $f : \boldsymbol{R}^{N^2} \rightarrow \boldsymbol{R}^{N^2}$. In this situation, each graph is represented by $N!$ different matrices which all have to be identified by the Transformer. This encoding also does not provide any high-level structural information about the graph.

**Naive Edge Transformer**. The naive edge Transformer has the same difficulty since the function its learning is $\boldsymbol{R}^{N(N-1)} \rightarrow \boldsymbol{R}^{N(N-1)}$ and the representation of each edge depend on a choice of labeling of the vertices and the $N!$ possible labelings need to be identified again.

**Node-LPE Transformer**. In the absence of eigenvalues with multiplicity $> 1$, the node LPE that learns a function $\boldsymbol{R}^{N \times 2 \times m} \rightarrow \boldsymbol{R}^{N \times k}$ does not take as input a representation of the graph that depends on the ordering of the nodes. It does, however, depend on the choice of the sign of each of the eigenvectors so there are still $2^N$ possible choices of graph representations that need to be identified by the Transformer but this is a big drop in complexity compared to the previous $N!$. The eigenfunctions also provide high-level structural information about the graph that can simplify the learning task of the graph.

**Edge-LPE Transformer**. Finally, the edge LPE of appendix A uses a graph representation as input that is also independent of the sign choice of the eigenvectors so each graph has a unique representation (considering the absence of eigenvalues with multiplicity $> 1$). Again, the eigenfunctions provide high-level structural information that is not available to the naive Transformer.

**LPE Transformers for non-isospectral graphs**. Isospectral graphs are graphs that have the same set of eigenvalues despite having different eigenvectors. Here, we argue that the proposed node LPE can approximate a solution to the graph isomorphism problem for all pairs of non-isospectral graphs, without having to learn invariance to the sign of their eigenvectors nor their multiplicities. By considering only the eigenvalues in the initial linear layer (assigning a weight of $0$ to all $\phi$), and knowing that the eigenvalues are provided as inputs, the model can effectively learn to replicate the input eigenvalues at its output, thus discriminating between all pairs of non-isospectral graphs. Hence, the problem of learning an invariant mapping to the sign of eigenvectors and multiplicities is limited only to non-isospectral graphs. Knowing that the ratio of isospectral graphs decreases as the number of nodes increases (and is believed to tend to 0) [35], this is especially important for large graphs and mitigates the problem of having to learn to identify $2^N$ with eigenvectors with different signs. In Figure 11, we present an example of non-isomorphic graphs that can be distinguished by their eigenvalues but not by the 1-WL test.

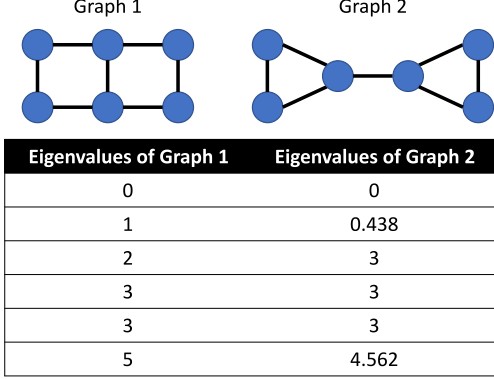

| Eigenvalues of Graph 1 | Eigenvalues of Graph 2 |
| --- | --- |
| 0 | 0 |
| 1 | 0.438 |
| 2 | 3 |
| 3 | 3 |
| 3 | 3 |
| 5 | 4.562 |

Figure 11: Example of non-isomorphic non-isospectral graphs that can be distinguished by the eigenvalues of their Laplacian matrix, but not by the 1-WL test.