# OpenReview forum: "Rethinking Graph Transformers with Spectral Attention"
_NeurIPS.cc/2021/Conference — NeurIPS 2021 Poster_

### Official Review · Reviewer_M4Qn · 2021-07-15

**Rating:** 6
**Confidence:** 3

**Summary:**

This work proposes a transformer designed for graph structured data. As opposed to GNNs, the transformer allows non-neighboring nodes to communicate. To avoid loosing information on the structure of the graph, the transformer needs some form of structure encoding of the nodes, a non trivial problem for graphs. A common PE scheme consists in using the spectrum of the graphs Laplacian but, as detailed by the authors, this raises some issues. The authors propose an improved positional encoding circumventing these problems and demonstrate that a fully connected transformer is competitive with GNNs on four datasets.

**Ethical Concerns:**

None.

**Limitations And Societal Impact:**

Both are adressed by the authors.

**Main Review:**

This work is one of the first suggesting that transformers equipped with structure encoding can rival with GNNs, which are the de facto architecture for graph-structured data. The paper seems sounds although some points could be clarified (see below).

Pros:
- The authors carefully detail the issues related to using the spectrum of the Laplacian as node positional encoding in a transformer, making interesting observations and proposing a solution.
- SAN performs experimentally well.

Cons:
- It is claimed in the main paper (l254) that the model is able to capture long-distance interactions, but as far as I understand there are no experiments supporting this.
- Do eigenvectors of different Laplacians (i.e of different graphs) belong to a similar space, hence can they really be "compared" as done in your method? What is your view on this?
- One or two more datasets would be welcomed to better understand how does SAN compare to GNNs, for example a small scale such as NCI1 and larger scale such as OGBG-MolPCBA.
- As mentioned by the authors, the sign flip is not avoided by the proposed method unless paying a price of $O(N^4)$ ($N$ being the number of nodes in the graph).

Questions and remarks:
- Wouldn't have been also possible to simply pad eigenvectors of the Laplacian and project the position encodings to a new space to respectively solve the "Variable number of eigenvectors problem" and "Multiplicities" problems?
- How is $m$ selected in practice?
- l155: minor typo but I am not sure we can talk about the "eigenfunction of a node".

**Time Spent Reviewing:**

3

---

> ### Author Response · Authors · 2021-08-08
> **Response to Reviewer M4Qn**
>
> We would like to thank you for your thorough review and hope to answer your concerns below.
>
> **Long-range dependencies**
>
> To the best of our knowledge, it is a difficult task on its own to quantify how well a model can capture long-range dependencies (there are no established experiments that directly measure it). However, we attempted to demonstrate its ability to do so with the ablation study where we saw how tuning the gamma parameter impacted results. Across all our experiments, using non-zero gamma (which increases the model’s ability to capture LR dependencies) improved results in specific ranges.
>
> Surprisingly, the gain in performance was most apparent on the CLUSTER dataset which has an intriguing long-range characteristic: the graphs are large (over 100 nodes) and a small sample of ~6 nodes have their true label as a node feature. We noticed immense gains when continuously increasing gamma, and we speculate that this is because each node in the graph becomes connected to these nodes with full connectivity.
> We would like to highlight that these points were addressed at the end of section 4 of the paper.
>
> **Eigenvector space**
>
> You raise a very important point. Are learned patterns in the eigenspace transferable between graphs?
> In the space of regular grids and sequences, there is no such concern since the eigenbasis are the same across grids. However, this does not hold with graphs. In fact, we added a section “2.3 Learning with the eigenfunctions” to directly answer your concern. Below is a summary of the points raised.
>
> - *Similar graphs have similar spectra*: Thus we can expect the network to be able to transfer some patterns across graphs by the similarity of their spectrum. In fact, spectral graph theory tells us that the lowest and largest non-zero eigenvalues are both linked to the geometry of the graph (algebraic connectivity and spectral radius).
>
> - *Relative position within each graph*: The proposed LPE directly compares the eigenfunctions of different nodes belonging to the same graph, not across graphs. This allows the model to understand both the absolute and relative position relative to other nodes within a given graph in the early layer. Hence, eigenspaces are used mainly to understand the relationship between nodes, not to compare graphs.
>
> - *The full eigenspace contains geometrical information*: Spectral graph theory has studied in depth the geometrical/physical properties of graphs from their Laplacian eigenfunctions. By developing a method that can use the full spectrum of a graph, the proposed model can theoretically capture this geometrical and physical information. Hence, our model does not aim to find differences between specific eigenvalues or eigenvectors of different graphs. Rather, it aims at finding differences between the full eigenspaces, which we theoretically know to be possible.
>
> **Datasets**
>
> As mentioned in the general comment addressed to all reviewers, we are aiming to provide results for the OGB-molPCBA dataset.
>
> **Question: Projecting the eigenvectors**
>
> In fact, projecting the eigenvectors is precisely what we sought to do. However, the process of projecting the eigenvectors is not trivial, and we cannot use simple methods like PCA or statistical pooling to accomplish this since all eigenvectors are indeed orthogonal.
> We decided to apply the Transformer to accomplish this, motivated by the sequential nature/ordering of the eigenvalues across the spectrum (see Figure 2). This further enables the model to dynamically learn the best projection during training.
>
> **Question: How is m selected in practice?**
>
> We have clarified this point in the paper to ensure that future readers know how to optimize it.
> The m is simply treated as a hyperparameter that can be tuned. We often relied on domain knowledge of the task to decide the range to try (i.e whether or not we expect the task to rely on long-range information).
> Furthermore, we expect smaller graphs and lower dimension graphs to require fewer frequencies since the absolute and relative positions of nodes can be learned with fewer parameters.
>
> **Thank you**
>
> We hope to have addressed your concerns, and we are happy to further discuss with you during the rebuttal period.

---

> > ### Comment · Reviewer_M4Qn · 2021-09-01
> > **Noted**
> >
> > Thank you for your detailed comment. I appreciate the discussion on the eigenvector space and think that this, combined with the more general discussion on the spectrum of the Laplacian, is a strong point of the paper. As mentioned by other reviewers, I look forward to see results on mol-pcba yet also more data constrained tasks. As for the assertion on long-range dependencies, why not trying to display learned motifs compared to other models (I will obviously not blame you for this as this is a late answer)?

---

### Official Review · Reviewer_FaY8 · 2021-07-16

**Rating:** 6
**Confidence:** 4

**Summary:**

This paper presents an interesting idea on designing a novel graph transformer with learning positional encoding by a spectral attention network (SAN) using the full Laplacian spectrum. Three theoretical limitations from previous efforts are overcome by this spectral attention, namely WL test and universality, over-squashing and physical interactions. Experimental results on four standard datasets demonstrate the effectiveness of the proposed SAN.

**Limitations And Societal Impact:**

Yes

**Main Review:**

Strengths:

S1: It is important to properly define the positions of Transformers on graph structured data.
S2: The proposed method on using the full spectrum of Laplacian as positional encoding is simple yet effective, and the complementary theoretical analysis on the better properties brought by using spectrum is interesting and rooted in spectral graph theory.
S3: The paper is easy to follow with a clear motivation and closely related to the NeurIPS community with comprehensive experimental analysis.

Weaknesses:

W1: The novel is rather limited. The idea that using the spectrum of Laplacian as positional encoding is already proposed in the GT model as the authors stated, and certain properties of SAN, such as WL test and universality and reduced over-squashing, seem like that can also be achieved by the GT model with slight change on the design of structure.

W2: One important design is that SAN considers the variable # of eigenvectors by using a hyperparameter $m$. However, the choice of $m$ that splits the spectrum into two parts based on a hyper parameter seems not well-supported. In my view, the definition of low/high frequency components should at least have something to do with the actual distribution of eigenvalues, and not simply be based on a hard threshold on the eigenvalue index. This should be carefully discussed in the paper.

W3: The reviews of previous works seem missing from both main paper and the Appendix. Some of the related works upon using the attention on the spectrum of Laplacian for both Transformer architecture [GTN and GT] and sparse message-passing models [1,2] should be discussed in details to demonstrate the novelty of proposed methods.

W4: For experiments, why there are so many missing results for baselines on MOLHIV dataset? This makes the role of this dataset a little meaningless. By the way, the statistics of datasets are missed. I noticed that the benchmarks used for evaluation are relatively small graphs, I would be curious about how the performance is on large graphs such as social networks.
[1] Dong, Y., Ding, K., Jalaian, B., Ji, S., & Li, J. (2021). Graph Neural Networks with Adaptive Frequency Response Filter. arXiv preprint arXiv:2104.12840.
[2] Chang, H., Rong, Y., Xu, T., Huang, W., Sojoudi, S., Huang, J., & Zhu, W. (2020). Spectral graph attention network. arXiv preprint arXiv:2003.07450.


**Time Spent Reviewing:**

14

---

> ### Author Response · Authors · 2021-08-08
> **Response to Reviewer FaY8**
>
> We would like to thank you for your thorough review and hope to answer your concerns below.
>
> **W1: The novelty is rather limited.**
>
> Despite the GT model being released a few months before our approach, our paper delivers novelty from **theoretical**, **engineering**, and **empirical** points of view. For the **theory**, we provide detailed explanations of the shortcomings of the positional encodings employed by the GT model and provide multiple proofs of the very strong expressivity of our model. In terms of **engineering**, our model uses a learned positional encoding that projects the full spectrum of the graph into a fixed size embedding, thus providing, for the first time, a method for defining absolute and relative positions of each node in a graph. Finally, from the **empirical** point of view, our method strongly outperforms the GT model and provides the first Transformer method that matches the SOTA of message passing. We believe that one should not attribute the novelty to GT, since despite giving the first hint of the idea, they were not successful in its implementation, and we provide much more than incremental improvements.
> We clarified the novelty of our paper in the introduction by highlighting differences between the GT model and our SAN model, which was summarized in Table 1.
>
> **W2: Variable number of eigenvectors**
>
> Although we believe that it is a theoretically good idea to select the m-first eigenfunctions depending on the distribution of eigenvalues, we did not explore this area since it has empirical problems. For instance, having no hard control on the number of eigenfunctions will cause the length of the sequences to be too different and will result in loss of memory due to over-padding in the batch of tensors. Secondly, defining such a function is a difficult task in itself and requires tuning on different datasets. We believe this task can be handled during training by the LPE module without hard-coding a function.
>
> **W3: Review of previous work**
>
> Thank you for proposing the works [1] and [2]. Since they are very recent, we missed them in our original literature review, but we have now added them with a mention of how they differ from our method. In summary, we noted the following:
> Recent work has also suggested using spectral properties to improve the performance of Attention for graph neural networks [1, 2]. However, a major difference is that we propose using Laplace eigenvector-based positional encodings as a soft inductive bias to enable fully-connected Attention over graphs. On the contrary, these works are limited to focusing on local Attention-based message passing. Respectively, they use the spectrum to define spectral filters [1] or wavelet filters [2] which are analogous to pure spectral methods, GAT, or even to DGN, *all of which are included in our original review*, and have stark differences from what we proposed.
> To answer your concern regarding the clear distinction between message-passing, GAT, and our model, we added a section in the appendix highlighting, both visually and technically, the architectural differences.
>
> **W4: Experimental evaluation**
>
> Regarding the sparsity of the models for the molHIV dataset, we simply provide the results for the models available on the online and public benchmark since they are the most reliable. Some models are missing, but note that the standard baseline of GNNs is present (GCN, GIN, GAT) and that the recent state-of-the-art is also present (PNA, DGN). Other models typically perform within their range.
> Regarding testing on social networks, unfortunately, this will cause a scaling problem with our model since fully connected Transformers do not scale well to large data. This could be solved in future work by only connecting close neighbors instead of the full graph.
> To improve on our experimental evaluation, we are committed to providing an additional benchmark for the molPCBA dataset, as mentioned in the general comment addressed to all Reviewers.
>
> **Thank you**
>
> We hope to have addressed your concerns, and we are happy to further discuss with you during the rebuttal period.

---

> > ### Comment · Reviewer_FaY8 · 2021-08-23
> > **Thank you for the Response**
> >
> > The authors addressed all my concerns except for W2. I am still curious the distribution of eigenvalues.  It is difficult for different datasets but it is still worth to try on one dataset as an ablation study in order to support the claim. I will raise my score.

---

### Official Review · Reviewer_rLqr · 2021-07-17

**Rating:** 8
**Confidence:** 3

**Summary:**

This paper proposes a learned position encoding mechanism which can fully make use of the eigenvalues and eigenvectors information from the graph Laplacian. The learned position encoding is then concatenated with the embedded node feature. In addition, the edge feature is also served as an input feature in order to completely leverage the graph structural information. Also, by fully connecting the graph, it helps tackle the common over-squashing problem in many GNNs. The paper also proposes an alternative version of attention so that real edges and added edges are taken into account separately when computing the attention. Eventually, the paper empirically tests the performance of the proposed model on real graph datasets.

**Limitations And Societal Impact:**

1.	As mentioned by the paper itself, a critical problem is the computational time complexity of the architecture, which requires $\mathcal{O}(N^{3})$ in order to generate node-wise embedding, or even $\mathcal{O}(N^{4})$ if the edge-wise embedding is leveraged.
2.	Even though the results achieved by the proposed model is impressive, I encourage the authors to also test more on other well-known graph benchmark datasets, for example the OGB dataset, which exploits a more challenging splitting procedure and is similar to real-life application, or some datasets mentioned in [1].

[1] Errica, Federico, et al. "A fair comparison of graph neural networks for graph classification." arXiv preprint arXiv:1912.09893 (2019).


**Main Review:**

1.	The innovative architecture can make use of the full spectral information, while many spectral GNNs only adopt the eigenvectors of the Laplacian. Section 2.1.2 mentions the intuitions behind using full spectral information, which come from Laplacian’s green function and node distance measures. I enjoy reading about this part.
2.	The paper also summarizes some classical problems encountered by methods based on eigenfunctions. The proposed method can tackle most of them, except for the sign invariance of eigenvectors. This issue can be further solved if the edge learned position encoding is used.


**Time Spent Reviewing:**

24

---

> ### Author Response · Authors · 2021-08-08
> **Response to Reviewer rLqr**
>
> We would like to thank you for your time spent reviewing the paper and are very happy with your positive feedback. We hope our paper can be the start of a long line of work for Transformers on graphs.
>
> **Datasets**
>
> As mentioned in the general comment addressed to all reviewers, we are currently working on adding empirical results for the OGB-molPCBA dataset.
>
> **Computational complexity**
>
> As mentioned in our paper, there has been a lot of recent work that reduces the computational complexity of Transformers. Due to limited resources, we did not address the issue in our current work, but it is easy to address in future work.

---

> > ### Comment · Reviewer_rLqr · 2021-09-01
> > **Thank you for the response**
> >
> > The authors mentioned that they will test the proposed model on the OGB-molPCBA. I look forward to the result and the improvement made on the paper. Since the authors also mentioned leveraging transformers with lower complexity helps reduce time complexity, it would be better if they can discuss more about it in the paper.

---

> > > ### Author Response · Authors · 2021-09-01
> > > **Preliminary results for OGB molPCBA**
> > >
> > > Dear reviewer rLqr,
> > >
> > > we have published our preliminary results for molPCBA in the general author comments. Our results are given below, with more details in the general comment. Note that we are still working on hyperparameter optimization.
> > >
> > > - Test: 27.65 +- 0.42
> > > - Val: 28.93 +- 0.16
> > >
> > > We compare our results to those reported in DGN [4]. Our method performs far better than GCN (20.20) and GIN (22.66), but slightly lower than the top reported models are PNA (28.38) and DGN (28.85).
> > >
> > > Regarding changes in the paper, we would like to clarify that we are not allowed to edit the PDF submitted to OpenReview during the rebuttal process. Rest assured that we will incorporate any promised change for the final submission.

---

### Official Review · Reviewer_kbAi · 2021-07-17

**Rating:** 6
**Confidence:** 4

**Summary:**

The central part of the paper is the design of a learned positional encoding (LPE) that can be used in graph transformers in a similar way the positional encoding is used in text transformers. The authors propose to use eigenvalues and eigenfunctions to encode node positions in a graph, and demonstrate how to incorporate them into a transformer model that achieves high expressivity. The architectural choices are based on the theoretical introduction in which some concepts are borrowed from physics, such as the analogy to electric potential between nodes. The LPE is designed to address five principles that were overlooked by different methods. The authors also implement a new model, Spectral Attention Network (SAN), that utilizes the LPE resulting from the theoretical findings. The experimental results confirm the efficacy of the method.

**Limitations And Societal Impact:**

The limitations and societal impacts are described in the paper.

**Main Review:**

The authors propose a novel learnable positional encoding and a new architecture, spectral attention network. The goal is to combat the problems of over-smoothing and over-squashing in graph neural networks and to increase expressivity. However, these problems are present in message passing or graph convolutional networks, and the authors do not elaborate on other graph transformers in this context in the work motivation. The correspondence between eigenvectors and sine functions is sketched in Section 2.1.1 An elegant physical explanation for the algorithmic decisions is provided. It is based on the Green's function and sheds new light on spectral graph theory and graph transformer architectures. The SAN architecture is an extension of the graph transformer proposed by Dwivedi and Bresson who also used Laplacian eigenvectors to encode node positions. However, their approach was much simpler and did not consider five principles raised by the authors of this paper -- they used $k$ smallest non-trivial vectors for each node.

The quality of the presented results is high. The performance metrics are calculated along with their error intervals based on 10 reruns of the model. The limitations of different elements of the architecture are described. The authors argue that their architecture is stronger than the Weisfeiler-Lehman isomorphism test and reduced over-squashing. However, there are some claims throughout the text that are not backed by any literature or experiments. For example in line 219: “Edge representations are not updated as it adds complexity with little to no performance gain”, or in line 254 "SAN model, which uses the Laplace spectrum more effectively, can learn to mimic the physical interactions presented in section 2.1.2". Is it confirmed by experiments?

Overall, the paper is clearly written and easy to follow. However I have some questions about Equations 1 and 2 since this is the source of the method motivation. Should not the Green's function be symmetric? $G$ is defined as the pseudo-inverse of the graph Laplacian, which is symmetric (for an undirected graph). By the definition of the generalized inverse, the inverse of the Laplacian would be
\begin{equation}
G(j_1, j_2) = \sum_{i>0} \frac{\phi_{i,j_1} \phi_{i,j_2}}{\lambda_i},
\end{equation}
but the derivation of Equation 1 is not immediately obvious for me. In [8] we see a similar formula (Equation 13), but this corresponds to the Laplace operator $\Delta$ which can be asymmetric, and even there the elements of $\phi_i$ are not squared. Am I missing something that is maybe related to the electric potential that was mentioned earlier? My other comments about the equations:
- In line 91, it probably should be said that $\hat{\phi}_{i,j}$ is the $j$-th **element** of the vector.
- $d_i$ is not defined.
- Should not the sum be over $i>0$ instead of $k>0$ in Equation 2?

The work is valuable from the perspective of the spectral graph theory research in deep learning. It can help to improve current graph transformer models. On the downside, only one graph transformer is used in the experiments. It would be interesting to see how SAN compares to the transformers that do not use the Laplacian eigenvectors as positional encoding. Additionally, in the caption of Figure 7, MoNet is mentioned as one of the architectures in the comparison, but the corresponding row is missing in the table. Another weak point of the experimental section is a relatively small range of tasks used. Maybe a task related to quantum chemistry would resonate well with the physical basis of the method.

Based on my above comments, I am leaning towards the rejection, but I am willing to change my rating if all the questions are addressed during the discussion period.

**Time Spent Reviewing:**

5

---

> ### Author Response · Authors · 2021-08-08
> **Response to Reviewer kbAi**
>
> We would like to thank you for your thorough review and hope to answer your concerns below.
>
> **Updating the edge layers**
>
> We added a mention in the paper that the choice of not updating the edge layer is empirical since we saw no benefits of using them during our early prototype. We also note that the SOTA methods, namely PNA and DGN, also do not update the edge representation across layers.
>
> **Learning physical interactions**
>
> We modified our statement to mention that it “can theoretically” learn any system based on these physical interactions, as supported by Appendix C3. Appendix C3 proves that the proposed model is a Universal approximator for graphs. It is also supported by intuition since the model can access the full spectrum of the Laplacian with information about the eigenvalues, which is directly related to the physical interactions of equations (1) & (2), and neglected by previous work.
> We added a mention that empirical validation of this statement is difficult to evaluate and that such evaluation is out of scope from our current work.
>
> **Green’s function**
>
> Thank you for noticing the mistake. You are right that there should not be a squared term. We further defined our Green’s function in regards to the symmetric Laplacian, which complicates the equation with degree terms $d_{j_1}, d_{j_2}$. We will use the definition you provided, which we also found in Fan Chung’s paper [8].
>
> **Graph Transformer in the experiments**
>
> Our paper already provides a comparison to GAT (a Transformer without positional encodings) in Figure 7. and an ablation study of the LPE in Figure 6, where we tested our method with and without the Laplacian positional encoding. As it wasn’t clear, we made it more explicit in the paper that these comparisons are available.
> Note that other absolute or relative positional encodings could be used, such as the distance to the centroid or the relative distance between each pair of nodes. However, at the time of submission, there existed no literature related to such encoding, and these ideas could be subject to another paper.
>
> **MoNet**
>
> Thank you for noticing, we have removed MoNet from the caption.
>
> **Quantum Chemistry Dataset**
>
> As mentioned in the global comment, we are currently running experiments on the OGB-molPCBA dataset and are committed to releasing the results prior to the camera-ready version.
>
> **Thank you**
>
> We hope to have addressed your concerns, and we are happy to further discuss with you during the rebuttal period.

---

> > ### Comment · Reviewer_kbAi · 2021-08-22
> > **Thank you for the response**
> >
> > Thank you for the reply. I think you addressed all my concerns, and I feel I understand the mathematical background better now. I have only one follow-up comment. As you pointed out, GAT is included in the experiments, but I do not think GAT should be called a transformer network. The attention mechanism is used there for weighting node neighbours. Probably this is a matter of definition, but transformers are typically designed to capture longer-range dependencies. To name a few examples of graph transformers:
> >
> > 1. GROVER [1] is a graph transformer for processing chemical data that operates on a fully connected graph with keys, queries, and values extracted by a graph neural network,
> > 2. Graph Transformer Networks (GTN) [2] is a transformer architecture that you described in your related work section,
> > 3. Graph-BERT [3] is another graph transformer; the transformer encoder processes linkless subgraphs, and different node embeddings are considered.
> >
> > I understand that [1] can be only applied to molecular graphs. However, it would be interesting to see how a general-purpose graph transformer compares to an architecture tailored to the molecular domain. In my opinion, including more transformer architectures is worth considering in the future.
> >
> > [1] Rong, Yu, et al. "Self-Supervised Graph Transformer on Large-Scale Molecular Data." Advances in Neural Information Processing Systems 33 (2020): 12559-12571.
> >
> > [2] Yun, Seongjun, et al. "Graph transformer networks." Advances in Neural Information Processing Systems 32 (2019): 11983-11993.
> >
> > [3] Zhang, Jiawei, et al. "Graph-BERT: Only attention is needed for learning graph representations." arXiv preprint arXiv:2001.05140 (2020).

---

> > > ### Author Response · Authors · 2021-08-26
> > > **Second response to Reviewer kbAi**
> > >
> > > We would like to thank you again for your thoughtful reviews and recommendations for our paper. We hope to address your remaining concern below and are happy to continue discussing with you regarding improving our paper.
> > >
> > > **Comparison to other Transformer-inspired GNNs**
> > >
> > > We omitted these papers in our original work because they lack official results on the main experimental benchmarks we considered (BGNN and OGB), and preferred to focus on the available results for their reliability. However, we agree it would be valuable to include some results from the papers you mentioned.
> > >
> > > Given the computational constraints required to fine-tune models properly, we can only assure that we will try our best to run some additional experiments for the camera-ready version.

---

### Author Response · Authors · 2021-08-08
**General Response**

We would like to thank all Reviewers for their constructive comments and suggestions on our work, “Rethinking Graph Transformers with Spectral Attention”. We will try to answer all of the concerns to the best of our abilities and are happy to further discuss any concerns thanks to the open format of the review.
A general note from most reviewers was the need for an additional benchmark. To answer this concern, we are currently benchmarking our model against the OGB-molPCBA dataset [19]. This dataset comprises 450k molecules with 128 classification labels from biological assays. Due to its large size, it takes around a full day for a single run on our GPUs. We are committed to providing preliminary results before the end of the rebuttal period and will include the final results in the camera-ready version of our paper.

---

### Author Response · Authors · 2021-09-01
**Preliminary results for OGB molPCBA**

Dear reviewers,

We are currently working on hyperparameter optimization for the OGB molPCBA dataset. Due to the large size of the dataset and the models, it can take up to 30 hours to train a single model, and our resources are limited.

Our preliminary results with 4 runs for the percentage of average precision-recall are:
Test: 27.65 +- 0.42
Val: 28.93 +- 0.16

When comparing to the state-of-the-art reported in the DGN paper [4], our model currently performs far better than GCN (20.20) and GIN (22.66). The top reported models are PNA (28.38) and DGN (28.85), slightly better than our reported performance.

Note that these results are still very interesting since they represent the first fully-connected graph Transformer to perform well. With our continued hyper-parameter search, we hope to achieve similar results to the SOTA for the final version.

A summary of the current top hyper-parameters is given below. The full set of parameters will be provided in the updated version of the anonymous repository

- Global
   - learning rate=0.0003
   - batch size=512
   - lr schedule patience=20
   - batch accumulation=2
   - dropout=0.5
- Main transformer:
   - depth=5
   - hidden dimension=304
   - number of heads=8
- Learned positional encoding
   - m=10
   - depth=3
   - hidden dimension=16

---

### Decision · Program_Chairs · 2021-09-27

**Decision:**

Accept (Poster)

**Comment:**

In this paper, a learned positional encoding (LPE) based on  eigenvalues and eigenvectors information from the graph Laplacian is designed to make Transformer work better on graph data. Experimental results show that the proposed method is effective on several important tasks. Most reviewers see the technical value of the paper, and agree that using graph Laplacian in the learning of positional encoding is a good idea. On the other hand, some concerns have been raised, mainly regarding the computational complexity and the completeness of the experimental results. The authors did a good job in providing their rebuttals, and after several rounds of discussions, the reviewers came to a consensus on the positive side. As a result, my recommendation is ACCEPT as a poster.